# Relationship between Leptin and Insulin Resistance among Community—Dwelling Middle-Aged and Elderly Populations in Taiwan

**DOI:** 10.3390/jcm11185357

**Published:** 2022-09-13

**Authors:** Yu-Lin Shih, Tzu-Cheng Huang, Chin-Chuan Shih, Jau-Yuan Chen

**Affiliations:** 1Department of Family Medicine, Chang-Gung Memorial Hospital, Linkou Branch, Taoyuan 333, Taiwan; 2United Safety Medical Group, General Administrative Department, New Taipei City 242, Taiwan; 3College of Medicine, Chang Gung University, Taoyuan 333, Taiwan

**Keywords:** HOMA-IR index, insulin resistance, leptin, middle-aged and elderly, metabolic syndrome, obesity

## Abstract

The relationship between leptin and insulin resistance among middle-aged and elderly populations in Asia is seldom reported. Our research included 398 middle-aged and elderly Taiwanese individuals. First, we divided participants into three groups according to the tertiles of the homeostasis model assessment of insulin resistance (HOMA-IR) to analyze the parameters between each group. Pearson’s correlation was then applied to calculate the correlation between HOMA-IR and cardiometabolic risk factors after adjusting for age. A scatter plot indicated a relationship between serum leptin levels and the HOMA-IR index. Finally, the coefficients of the serum leptin level and HOMA-IR were assessed by multivariate linear regression. The participants in the high HOMA-IR index group were more likely to have higher serum leptin levels. Meanwhile, the HOMA-IR index was positively correlated with serum leptin levels, even after adjusting for age. Serum leptin levels were positively correlated with the HOMA-IR index (β = 0.226, *p* < 0.01) in the multivariate linear regression after adjusting for age, sex, smoking, drinking, BMI, triglycerides, systolic blood pressure, fasting plasma glucose, uric acid, ALT, and creatinine. Furthermore, the leptin–creatinine ratio also showed a significantly positive relationship with HOMA-IR in the same multivariate linear regression model. In conclusion, serum leptin levels showed a positive relationship with insulin resistance in middle-aged and elderly people in Taiwan. Furthermore, serum leptin levels may be an independent risk factor for insulin resistance according to our study.

## 1. Introduction

Insulin resistance relates to the accumulation of adipocytes in metabolic disorders, although the mechanism is not fully understood. The large amounts of data from previous human studies and animal research indicate a positive relationship between insulin resistance and metabolic disorders [1]. Furthermore, insulin resistance serves as the key feature of these metabolic disorders, such as diabetes mellitus (DM). Insulin resistance decreases the sensitivity of the physiological function of insulin and leads to type 2 DM [2,3]. DM-related complications carry serious consequences, including nephropathy, retinopathy, peripheral neuropathy, end-stage renal disease, cardiovascular disease, cerebrovascular disease, and diabetic peripheral circulatory disorder [4,5,6,7,8]. Recent studies even indicated that insulin resistance relates to other diseases, such as hypertension, cancer, and autoimmune diseases [9,10,11]. Diseases related to insulin resistance have created a significant socioeconomic burden for patients and society.

Leptin is believed to act as a negative feedback signal that regulates the energy balance by inhibiting food intake, and it reduces fat storage [12]. Leptin is mainly produced by adipocytes, and the plasma leptin level correlates with the number of adipocytes in the body [13]. Excessive fat tissue does not merely elevate plasma leptin levels but relates to insulin resistance [14], which is a crucial feature of metabolic disorders. Moreover, leptin resistance was also found in the obese population [15], who are vulnerable to insulin resistance. Hence, we aimed to investigate the relationship between leptin levels and insulin resistance in the population aged between 50 and 85 years in Taiwan. The results of our study may provide a possible reference for insulin resistance in primary care.

## 2. Materials and Methods

### 2.1. Study Population

This retrospective community-based study was performed using data acquired from a community health survey program in Northern Taiwan in 2019. The participants were included based on the following criteria: aged between 50 and 85 years, living in the community, ability to complete a questionnaire, and completion of all examinations. Some were excluded based on the following criteria: inability to complete all examinations, a history of recent heart disease, and those who declined to participate. Finally, 396 participants were enrolled in the study and qualified for analysis. In this study, a questionnaire that included medical history and personal information was administered. The questionnaire was completed during the interviews. Informed consent was obtained from all participants before enrollment. The study protocol was approved by the Institutional Review Board of the Linkou Chang Gung Memorial Hospital (IRB, No. 201801803B0).

### 2.2. Data Collection and Measurements

The content of the questionnaire included age, gender, current smoking status (current smoker or not), and alcohol drinking status (drinking on more than two days per week or not). Smoking and drinking status were self-reported. The health survey collected data on hyper-tension (HTN), DM, and dyslipidemia. Systolic blood pressure (SBP, mmHg) and diastolic blood pressure (DBP, mmHg) were measured at rest more than two times. BMI was calculated as the person’s weight in kilograms divided by the square of their height in meters. Waist circumference (WC; cm) was measured at the midpoint between the inferior margin of the last rib and the iliac crest in the horizontal plane in an upright position. The biochemical laboratory data were analyzed at the Roche model lab at the Taiwan E&Q Clinical Laboratory, including leptin levels (ng/mL), fasting plasma glucose (FPG, mg/dL), triglyceride levels (TG, mg/dL), high-density lipoprotein (HDL-C, mg/dL), low-density lipoprotein (LDL-C, mg/dL), albumin (g/dL), alanine transaminase (ALT, mg/dL), creatinine (mg/dL), uric acid (mg/dL), and insulin (µU/mL).

### 2.3. Assessment of Insulin Resistance

In this study, insulin and FPG levels were measured at the Roche model lab at the Taiwan E&Q Clinical Laboratory. The participants were categorized according to the tertiles of the HOMA-IR into low, middle, and high HOMA-IR level groups. HOMA-IR was calculated using the following formula: [glucose (mg/dL) × insulin (mIU/L)]/405.

### 2.4. Definition of HTN, DM, and Dyslipidemia

HTN was defined as SBP ≥ 140 mmHg, DBP ≥ 90 mmHg, a history of hypertension, or treatment for HTN [16]. DM was defined as an FPG level ≥ 126 mg/dL, DM history, insulin therapy, or oral antidiabetic drugs [17]. Dyslipidemia was defined as TG ≥ 150 mg/dL, TC ≥ 200 mg/dL, LDL-C ≥ 130 mg/dL, HDL-C < 40 mg/dL in men or <50 mg/dL in women, dyslipidemia history, or the use of lipid-lowering medication or underlying disease [18].

### 2.5. Statistical Analysis

The enrolled individuals were divided into three groups according to the HOMA-IR index levels: low, middle, and high. For the laboratory and anthropometric data of each group, categorical variables were expressed as *n* (%) and analyzed using the chi-square test. We checked the normality of the continuous variables by using the Shapiro–Wilk normality test. We presented the data as the mean ± [SD] if our data were consistent with normally distributed variables and as the median (Q1, Q3) if the variables (leptin, HOMA-IR, FPG, insulin, ALT, triglyceride, HDL-C, creatinine) significantly deviated from a normal distribution, as shown in Table 1. *p*-values were derived from one-way ANOVA for data consistent with a normal distribution and from Kruskal–Wallis ANOVA for data consistent with a non-normal distribution. Pearson’s correlation coefficient was used to analyze the correlations between the HOMA-IR index and age, leptin, FPG, insulin, ALT, HDL-C, LDL-C, triglycerides, uric acid, creatinine, SBP, waist circumference, and BMI. Pearson’s correlation adjusted for age was also arranged. Finally, linear regression analysis was used, with leptin as an independent factor, to evaluate the association between leptin and HOMA-IR index level. Three linear regression models were used in this study. Model 1 was unadjusted; model 2 was adjusted for age, gender, smoking, drinking, and BMI; and model 3 was adjusted for age, sex, smoking, drinking, BMI, triglycerides, systolic blood pressure, fasting plasma glucose, uric acid, alanine aminotransferase, and creatinine levels. Then, the same linear regression model was performed again to evaluate the relationship between the leptin–creatinine ratio and insulin resistance. Statistical significance was set at *p* < 0.05. All statistical analyses were performed using SPSS for Windows (IBM Corp. Released 2011. IBM SPSS Statistics, version 20.0. Armonk, NY, USA)

## 3. Results

This study was conducted on elderly and middle-aged people from communities in Northern Taiwan. A total of 396 individuals, including 164 men (41.4%) and 232 women (58.6%) with a mean age of 63.72 ± 8.76 years, were enrolled for analysis. The median of the HOMA-IR index in the study group was 2.50 [1.50, 4.10]. Table 1 summarizes the anthropometric data of the study participants. The enrolled participants were divided into three groups according to the three tertiles of the HOMA-IR index. The lower tertile was defined as a HOMA-IR index < 1.8. The middle tertile was defined as a HOMA-IR index between 1.8 and 3.5. The upper tertile was defined as a HOMA-IR index > 3.5. A one-way ANOVA, Kruskal–Wallis ANOVA, and chi-square test were used to compare the clinical and anthropometric data in different groups. There was no statistically significant difference in age, LDL-C, albumin, and drinking status between the different HOMA-IR index groups. However, the participants in the high HOMA-IR group were more likely to have a higher leptin level. In addition, the participants in the high HOMA-IR group were more likely to be men. The participants in the high HOMA-IR group tended to have chronic diseases, such as hypertension, DM, and dyslipidemia. Higher FPG, insulin levels, triglycerides, uric acid, ALT, creatinine, WC, SBP, DBP, BMI, and smoking rates were also found in the high leptin group. HDL-C levels were lower in the high HOMA-IR group.

The correlations between HOMA-IR and various indicators of cardiometabolic risk are shown in Table 2. HOMA-IR was positively correlated with leptin levels, with statistical significance (*p* < 0.001). Additionally, HOMA-IR was positively correlated with FPG, insulin, ALT, triglycerides, uric acid, SBP, WC, and BMI. The positive correlation between these indicators and HOMA-IR remained significant even after adjusting for age. An inverse correlation was found between HOMA-IR and HDL-C levels, with statistical significance. There was no significant correlation between HOMA-IR index and LDL-C, creatinine, or age.

Figure 1 presents a scatterplot of leptin levels according to the HOMA-IR index. The trend line is also shown in Figure 1. A positive correlation between HOMA-IR and leptin levels is revealed by the fit line. Pearson’s correlation coefficient was 0.233, with a *p*-value < 0.001.

As shown in Table 3, three linear regression models were used to investigate the relationship between leptin levels and the HOMA-IR index. Model 1 was unadjusted; model 2 was adjusted for age, gender, smoking, drinking, and BMI; and model 3 was adjusted for age, gender, smoking, drinking, BMI, triglycerides, SBP, FPG, uric acid, ALT, and creatinine. The regression coefficient revealed a statistically significant positive correlation between HOMA-IR and leptin levels in all three models. The beta coefficient was 0.233 in model 1, 0.208 in model 2, and 0.226 in model 3.

We used the same multivariate model in Table 3 to evaluate the relationship between the leptin–creatinine ratio in Table 4. The relationships still remained significantly positive, with a beta coefficient of 0.037 in model 1, 0.031 in model 2, and 0.041 in model 3.

## 4. Discussion

By looking at the tertiles of the HOMA-IR index in Table 1, it can be seen that there was an increase in the FPG, insulin, and triglyceride levels as the HOMA-IR index increased. These results corresponded with those of previous studies, which indicated that HOMA-IR is positively correlated with metabolic factors [19,20,21,22]. Previous research indicated that HDL has an inverse relationship with HOMA-IR [20], which was also observed in our study. HOMA-IR is a risk factor for metabolic syndrome [19,22]. Moreover, we found an increase in SBP, DBP, BMI, WC, HTN, DM, and dyslipidemia in the high HOMA-IR index group. Dysfunction of the kidney and liver was also reported in those with insulin resistance [23,24,25], and we also found elevated levels of uric acid, ALT, and creatinine in the high HOMA-IR index group. Insulin is degraded by insulin-degrading enzymes in the liver [26]. The liver function has a profound impact on insulin degradation [27]. Indeed, there was a significant relationship between HOMA-IR and ALT, as shown in Table 1, and Pearson’s correlation also indicated that HOMA-IR had a significant link with ALT, even after adjustment for age. However, HOMA-IR still maintained a significant relationship with serum leptin levels, even after adjustment for ALT. Dehydration and hypoalbuminemia are often noted in older people [28,29]. Additionally, serum albumin levels also serve as an important indicator of liver function [30]. In our study, the average serum albumin levels of each group were all within the normal range, and there was no relationship between albumin and HOMA-IR in our study. Male sex and smoking status also had a positive relationship with HOMA-IR, which was also found in other studies [31,32]. Meanwhile, increasing leptin levels are directly associated with a high HOMA-IR index. These findings led us to speculate that an association exists between leptin levels and other confounding factors.

As shown in Table 2, we found a positive correlation between HOMA-IR and leptin levels, along with other risk factors for insulin resistance, such as triglycerides, FPG, insulin levels, WC, and BMI. HDL is a protective factor against insulin resistance [20], which is in line with our results. The correlation of these factors reached statistical significance, even after adjustment for age. Pearson’s correlation between HOMA-IR and leptin levels was 0.241, with a *p* < 0.001 after the adjustment for age. Previous research suggested that elevated leptin levels play an important role in insulin resistance [33]. Figure 1 shows a scatterplot of leptin levels according to the HOMA-IR index, and an obvious positive correlation between leptin levels and insulin resistance can be noted. Our results indicated a strong relationship between leptin levels and insulin resistance. Therefore, we wished to determine whether leptin levels could be an independent risk factor for predicting insulin resistance.

Table 3 shows the results of the linear regression analysis; the result maintained statistical significance from the unadjusted model to the adjusted model. After adjusting for age, sex, smoking and drinking status, BMI, triglycerides, SBP, FPG, uric acid, ALT, and creatinine, the unstandardized regression coefficient (B) was 0.226, and the standard deviation was 0.015. Because many low-molecular-weight proteins such as leptin are cleared by the kidneys, renal function plays a crucial role in the clearance of these proteins [34]. Creatinine was further taken into consideration in Table 4 for the linear regression. In Table 4, we used the same linear regression model in Table 3 to evaluate the relationship between the leptin–creatinine ratio and HOMA-IR IN in Table 4. The result revealed a significantly positive relationship in all three models. The linear regression analysis confirmed that leptin was an independent risk factor for insulin resistance, even considering the creatinine level.

Leptin changes our view of adipocytes as passive containers for fat to active endocrine tissues that mediate metabolism [35]. Leptin is mainly secreted from adipose tissue and is a 14 kDa polypeptide encoded by the ob gene [36]. Leptin mediates PI3K signaling, which is important for the regulation of glucose metabolism [37]. Leptin can also modulate JAK-STAT signaling in the hypothalamus to regulate body weight and food intake [38]. Leptin inhibits the accumulation of triglycerides in the muscle and liver by modulating the function of AMPK [39]. Furthermore, leptin regulates pancreatic β-cell function [38]. Overall, leptin decreases food intake and enhances insulin function. Leptin resistance is indicated by elevated leptin levels, as observed in obese individuals without an adequate leptin-mediated response [40]. A previous study observed an impairment in leptin transportation across the blood–brain barrier in obese rodents [41], and the decrease in leptin levels in the hypothalamus compromised the function of leptin, including insulin sensitivity [36]. In our study, we analyzed many metabolic parameters to further evaluate the relationship between HOMA-IR and leptin levels, which is an important indicator of insulin resistance. After we considered all the metabolic parameters in our study, the results showed that leptin levels can be an independent risk factor for HOMA-IR.

Our study also has epidemiological importance. Due to the development of China, the population of the country has developed healthier lifestyles and has greater longevity than before [42]. Obesity among the elderly population has emerged as a health problem [43]. When we compared the Chinese population with the Taiwanese population in our study, the indicators of metabolic syndrome, such as BMI and the rate of DM, were higher in the Taiwanese elderly population than they were in the Chinese counterpart. Our study can provide insight into the future elderly population of China.

There are several strengths of our study. First, our study had a straightforward design, a sufficient sample size, comprehensive and relevant confounders, and appropriate data analysis. Second, a novelty of our study was in discovering the strong relationship between leptin and insulin resistance in a community-dwelling population; there is no similar research that investigates this topic in the middle-aged and older populations in Taiwan. Third, our study has epidemiological importance, as it provides a possible prediction of the future elderly population in China. However, there are some limitations to our study. We recruited participants from Northern Taiwan as the target population. These participants might differ from the general population, and selection bias should be considered; the findings cannot represent the entire middle-aged and elderly population in Taiwan. Although our study revealed obvious results, future studies using the random sampling of communities with a wider range of regions should be considered. Moreover, other confounders, such as medication and hepatic steatosis, were not considered in our study. These potential confounders should be included in future studies.

## 5. Conclusions

This study showed that increased leptin levels are associated with high HOMA-IR, which is an important index of insulin resistance among middle-aged and elderly populations in Taiwan. Individuals with high leptin levels should be monitored in this age group, and adequate healthcare should be provided to prevent chronic diseases related to insulin resistance.

## Figures and Tables

**Figure 1 jcm-11-05357-f001:**
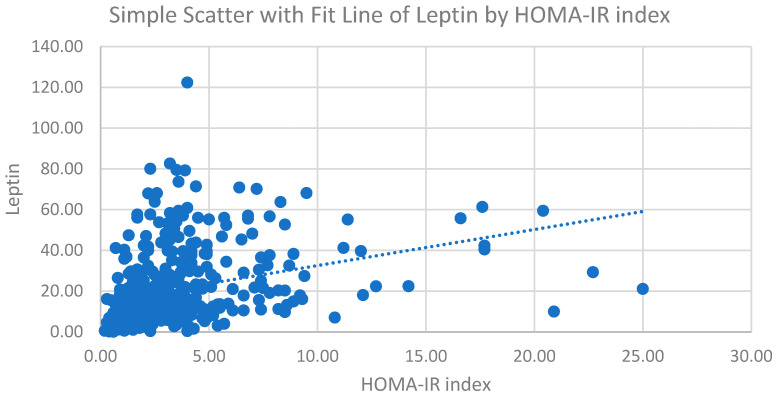
The correlation between leptin and the HOMA-IR index.

**Table 1 jcm-11-05357-t001:** Characteristics of participants according to the tertiles of the HOMA-IR index.

	HOMA-IR				
	Total	Low (<1.8)	Middle (1.8~3.5)	High (>3.5)	
Variable	N = 396	n = 129	n = 136	n = 131	*p* Value
Leptin (ng/mL)	14.89 (8.48, 28.36)	9.96 (5.72, 16.15)	14.30 (7.81, 26.55)	23.30 (13.24, 41.21)	<0.001
HOMA-IR index	2.50 (1.50, 4.10)	1.20 (0.90, 1.50)	2.50 (2.10, 3.08)	5.10 (4.10, 7.70)	<0.001
Age (years)	63.72 ± 8.76	63.90 ± 9.71	63.89 ± 8.13	63.36 ± 8.45	0.620
FPG (mg/dL)	99.00 (89, 118.75)	91.00 (84.00, 97.00)	98.50 (90.25, 111.00)	118.00 (105.00, 145.00)	<0.001
Insulin (uU/mL)	9.89 (6.17, 14.93)	5.16 (3.96, 6.44)	9.99 (8.49, 11.89)	18.22 (13.91, 24.86)	<0.001
Uric Acid (mg/dL)	5.63 ± 1.52	5.25 ± 1.30	5.48 ± 1.39	6.18 ± 1.70	<0.001
ALT (U/L)	21.00 (16.00, 30.00)	18.00 (15.00, 26.50)	20.00 (15.00, 25.00)	26.00 (19.00, 38.00)	<0.001
Triglyceride (mg/dL)	118.00 (86.00, 165.00)	92.00 (73.00, 120.50)	125.50 (88.25, 167.75)	141.00 (106.00, 205.00)	<0.001
HDL-C (mg/dL)	52.00 (43.00, 61.00)	58.00 (50.00, 68.50)	51.00 (43.00, 59.75)	47.00 (40.00, 56.00)	<0.001
LDL-C (mg/dL)	109.69 ± 33.99	111.97 ± 32.56	113.38 ± 37.06	103.62 ± 31.38	0.047
Creatinin (mg/dL)	0.80 (0.68, 0.97)	0.77 (0.66, 0.94)	0.82 (0.68, 1.00)	0.84 (0.73, 0.98)	0.027
Albumin (g/dL)	4.43 ± 0.25	4.43 ± 0.28	4.44 ± 0.26	4.42 ± 0.22	0.731
SBP (mmHg)	137.30 ± 17.49	134.58 ± 18.28	136.68 ± 15.96	140.63 ± 17.81	0.005
DBP (mmHg)	85.19 ± 10.98	82.38 ± 10.52	86.09 ± 11.06	87.02 ± 10.87	<0.001
BMI (kg/m^2^)	25.59 ± 3.84	23.46 ± 2.88	25.37 ± 3.05	27.92 ± 4.11	<0.001
WC (cm)	85.36 ± 10.83	82.81 ± 9.85	84.08 ± 10.00	89.17 ± 11.58	<0.001
Gender (male)	164 (41.4%)	42 (32.6%)	59 (43.4%)	63 (48.1%)	0.011
Smoking	50 (12.6%)	10 (7.8%)	17 (12.5%)	23 (17.6%)	0.017
Drinking	28 (7.1%)	8 (6.2%)	10 (7.4%)	10 (7.6%)	0.653
HTN	201 (50.8%)	43 (33.3%)	74 (54.4%)	84 (64.1%)	<0.001
DM	133 (33.6%)	25 (19.4%)	41 (30.1%)	67 (51.1%)	<0.001
Dyslipidemia	153 (38.6%)	36 (27.9%)	53 (39.0%)	64 (48.9%)	0.001

Note: Data are expressed as the mean ± (SD) for continuous variables with a normal distribution, as the median (Q1, Q3) for continuous variables that significantly deviated from a normal distribution, and as n (%) for categorical variables. HOMA-IR, Model Assessment-Insulin Resistance index; ALT, alanine transaminase; BMI, body mass index; DBP, diastolic blood pressure; DM, diabetes mellitus; FPG, fasting plasma glucose; HDL-C, high-density lipoprotein; HTN, hypertension; LDL-C, low-density lipoprotein; SBP, systolic blood pressure; WC, waist circumference.

**Table 2 jcm-11-05357-t002:** Pearson’s correlation coefficients between the HOMA-IR index and cardiometabolic factors.

	HOMA-IR Index			
	Unadjusted		Adjusted for Age	
Variables	Correlation	*p* Value	Correlation	*p* Value
Age (year)	−0.053	0.295	NA	NA
Leptin (ng/mL)	0.233	<0.001	0.241	<0.001
FPG (mg/dL)	0.436	<0.001	0.436	<0.001
Insulin (uU/mL)	0.935	<0.001	0.935	<0.001
ALT (U/L)	0.330	<0.001	0.329	<0.001
HDL-C (mg/dL)	−0.207	<0.001	−0.205	<0.001
LDL-C (mg/dL)	−0.075	0.136	−0.088	0.080
Triglyceride (mg/dL)	0.125	0.013	0.121	0.017
Uric Acid (mg/dL)	0.143	0.004	0.144	0.004
Creatinine (mg/dL)	0.072	0.154	0.084	0.097
SBP (mmHg)	0.109	0.030	0.124	0.014
WC (cm)	0.319	<0.001	0.323	<0.001
BMI (kg/m^2^)	0.322	<0.001	0.320	<0.001

Note: HOMA-IR, Model Assessment-Insulin Resistance index; ALT, alanine transaminase; BMI, body mass index; DBP, diastolic blood pressure; DM, diabetes mellitus; FPG, fasting plasma glucose; HDL-C, high-density lipoprotein; HTN, hypertension; LDL-C, low-density lipoprotein; SBP, systolic blood pressure; WC, waist circumference.

**Table 3 jcm-11-05357-t003:** Linear regression between leptin and the HOMA-IR index.

	Model 1			Model 2			Model 3		
	β	S.E.	*p* Value	β	S.E.	*p* Value	β	S.E.	*p* Value
Leptin	0.233	0.012	<0.001	0.208	0.017	0.002	0.226	0.015	<0.001

Model 1 was unadjusted. Model 2 was adjusted for age, gender, smoking, drinking, and BMI. Model 3 was adjusted for age, gender, smoking, drinking, BMI, triglyceride, systolic blood pressure, fasting plasma glucose, uric acid, ALT, and creatinine levels.

**Table 4 jcm-11-05357-t004:** Linear regression between the leptin–creatinine ratio and the HOMA-IR index.

	Model 1			Model 2			Model 3		
	β	S.E.	*p* Value	β	S.E.	*p* Value	β	S.E.	*p* Value
Leptin	0.037	0.010	<0.001	0.031	0.014	0.028	0.041	0.013	0.001

Model 1 was unadjusted. Model 2 was adjusted for age, gender, smoking, drinking, and BMI. Model 3 was adjusted for age, sex, smoking, drinking, BMI, triglyceride, systolic blood pressure, fasting plasma glucose, uric acid, and ALT level.

## Data Availability

The raw data supporting the conclusions of the article will be made available by the authors, without undue reservation.

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
