# Peer review of "Relationship between Leptin and Insulin Resistance among Community—Dwelling Middle-Aged and Elderly Populations in Taiwan"

_jcm, 2022, doi:10.3390/jcm11185357_

Round 1

Reviewer 1 Report

-        The study lacks novelty.

-        The Introduction section should be extended with more information about leptin and its influence on metabolic disorders.

-        Was medications use included in the statistical analysis as the confounding factor? This should be clearly stated.

-        In Statistical analysis sub-section the Authors should clearly explain how the distribution of data was checked. Which test did the Authors use?

-        The continuous data without normal distribution (such as HOMA-IR) should be presented as median (interquartile range).

-        The Authors in the Discussion section should not repeat the results from the previous section. Instead, the obtained results should be discussed and compared with the results of the previous studies related to this topic.

-        What are the limitations of the study?

-       Author Response

***Reviewer 1***

Question 1

-     The study lacks novelty

Answer 1

We thank the reviewer for reminding us this issue and our study still has some strengths. In response to this comment, we’ve added this to the discussion: ” There are several strengths of our study. First, our study had a straightforward design, sufficient sample size, comprehensive and relevant confounders, and appropriate data analysis. Second, a novelty of our study was in discovering the strong relationship between leptin and insulin resistance in a community-dwelling population; there is no similar research that investigates this topic in the middle-aged and older populations in Taiwan. Third, our study has epidemiological importance as it provides a possible prediction of the future elderly population in China.”

Question 2

-     The Introduction section should be extended with more information about leptin and its influence on metabolic disorders.

      Answer 2

Thank you for the accurate comments about this issue. We revised the Introduction as below:

“Insulin resistance relates to the accumulation of adipocytes in metabolic disorders, although the mechanism is not fully understood. The large amounts of data from previous hu-man studies and animal research indicate a positive relationship between insulin resistance and metabolic disorders [1]. Furthermore, insulin resistance serves as the key feature of these metabolic disorders, such as diabetes mellitus (DM). Insulin resistance decreases the sensitivity of the physiological function of insulin and leads to type 2 DM [2] [3]. DM-related complications carry serious consequences, including nephropathy, retinopathy, peripheral neuropathy, end-stage renal disease, cardiovascular disease, cerebrovascular disease, and diabetic peripheral circulatory disorder [4] [5] [6] [7] [8]. Recent studies even indicated that insulin resistance relates to other diseases, such as hypertension, cancer, and autoimmune diseases [9] [10] [11]. Diseases related to insulin resistance have created a significant socioeconomic bur-den for patients and society.

Leptin is believed to act as a negative feedback signal that regulates the energy balance by inhibiting food intake, and it reduces fat storage [12]. Leptin is mainly produced by adipocytes, and the plasma leptin level correlates with the number of adipocytes in the body [13]. Excessive fat tissue does not merely elevate plasma leptin levels but relates to insulin resistance [14], which is a crucial feature of metabolic disorders. Moreover, leptin resistance was also found in the obese population [15], who are vulnerable to insulin resistance. Hence, we aimed to investigate the relationship between leptin levels and insulin resistance in the population aged between 50 and 85 years in Taiwan. The results of our study may provide a possible reference for insulin resistance in primary care. ''

Reference:

  1. Blüher, Matthias. "Adipose tissue inflammation: a cause or consequence of obesity-related insulin resistance?." Clinical science 130.18 (2016): 1603-1614.
  2. Freeman AM, Pennings N. Insulin Resistance. 2021 Jul 10. In: StatPearls [Internet]. Treasure Island (FL): StatPearls Publishing; 2021 Jan–. PMID: 29939616.
  3. Kahn BB, Flier JS. Obesity and insulin resistance. J Clin Invest. 2000 Aug;106(4):473-81. doi: 10.1172/JCI10842. PMID: 10953022; PMCID: PMC380258.
  4. Ferrannini E, Natali A, Capaldo B, Lehtovirta M, Jacob S, Yki-Järvinen H. Insulin resistance, hyperinsulinemia, and blood pressure: role of age and obesity. European Group for the Study of Insulin Resistance (EGIR). Hypertension. 1997 Nov;30(5):1144-9. doi: 10.1161/01.hyp.30.5.1144. PMID: 9369268.
  5. L.N. Tseng, Y.H. Tseng, Y.D. Jiang, C.H. Chang, C.H. Chung, B.J. Lin, et al. Prevalence of hypertension and dyslipidemia and their associations with micro- and macrovascular diseases in patients with diabetes in Taiwan: an analysis of nationwide data for 2000-2009. J Formos Med Assoc, 111 (11) (2012), pp. 625-636
  6. H.Y. Li, Y.D. Jiang, C.H. Chang, C.H. Chung, B.J. Lin, L.M. Chuang. Mortality trends in patients with diabetes in Taiwan: a nationwide survey in 2000-2009. J Formos Med Assoc, 111 (11) (2012), pp. 645-650
  7. Y.J. Sheen, T.C. Li, J.L. Lin, W.C. Tsai, C.D. Kao, C.T. Bau, et al. Association between thermal threshold abnormalities and peripheral artery disease in patients with type 2 diabetes. Medicine (Baltimore), 97 (51) (2018), Article e13803
  8. Y.J. Sheen, W.H. Sheu. Association between hypoglycemia and dementia in patients with type 2 diabetes. Diabetes Res Clin Pract, 116 (2016), pp. 279-287
  9. C.L. Hsu, W.H. Sheu. Diabetes and shoulder disorders. J Diabetes Investig, 7 (5) (2016), pp. 649-651
  10. Orgel E, Mittelman SD. The links between insulin resistance, diabetes, and cancer. Curr Diab Rep. 2013 Apr;13(2):213-22. doi: 10.1007/s11892-012-0356-6. PMID: 23271574; PMCID: PMC3595327.
  11. Chung CP, Oeser A, Solus JF, Gebretsadik T, Shintani A, Avalos I, Sokka T, Raggi P, Pincus T, Stein CM. Inflammation-associated insulin resistance: differential effects in rheumatoid arthritis and systemic lupus erythematosus define potential mechanisms. Arthritis Rheum. 2008 Jul;58(7):2105-12. doi: 10.1002/art.23600. PMID: 18576352; PMCID: PMC2755593.
  12. Kelesidis T, Kelesidis I, Chou S, Mantzoros CS. Narrative review: the role of leptin in human physiology: emerging clinical applications. Ann Intern Med. 2010 Jan 19;152(2):93-100. doi: 10.7326/0003-4819-152-2-201001190-00008. PMID: 20083828; PMCID: PMC2829242.
  13. Schwartz, Michael W. "Central nervous system regulation of food intake." Obesity 14.2S (2006): 1S.
  14. Osegbe I, Okpara H, Azinge E. Relationship between serum leptin and insulin resistance among obese Nigerian women. Ann Afr Med. 2016 Jan-Mar;15(1):14-9. doi: 10.4103/1596-3519.158524. PMID: 26857932; PMCID: PMC5452686.
  15. Smith U, Kahn BB. Adipose tissue regulates insulin sensitivity: role of adipogenesis, de novo lipogenesis and novel lipids. J Intern Med. 2016 Nov;280(5):465-475. doi: 10.1111/joim.12540. Epub 2016 Oct 3. PMID: 27699898; PMCID: PMC5218584.
  16. Enriori, P. J., Evans, A. E., Sinnayah, P., & Cowley, M. A. (2006). Leptin resistance and obesity. Obesity14(S8), 254S-258S.

Question 3

-     Was medications use included in the statistical analysis as the confounding factor? This should be clearly stated.

Answer 3

Thank you for the accurate comments about this issue. Our study did not include the medication of the participants as the confounding factor because many participants also take the prescription from other clinics, and they do not know the prescription clearly. Further investigation between leptin level and medication should be considered in future studies. So we state this in our limitation as below “ Moreover, other confounders, such as medication, hepatic steatosis, and other low-molecular-weight proteins, were not considered in our study. These potential confound-ers should be included in future studies.”

Question 4

-     In Statistical analysis sub-section the Authors should clearly explain how the distribution of data was checked. Which test did the Authors use?

Answer 4

Thank you for this valuable comment, and we revised the related paragraph below “ We checked the normality of the continuous variables by using the Shapiro–Wilk normality test. We presented data as mean ± [SD] if our data were consistent with normally distributed variables, and median [Q1, Q3] if the variables (leptin, HOMA-IR, FPG, insulin, ALT, tri-glyceride, HDL-C, creatinine) significantly deviated from a normal distribution, as shown in Table 1. p-values were derived from one-way ANOVA for data consistent with a normal dis-tribution and Kruskal–Wallis ANOVA for data consistent with a non-normal distribution.” The corresponded content has edited as well. 

Question 5

-    The continuous data without normal distribution (such as HOMA-IR) should be presented as median (interquartile range).

Answer 5

We thank reviwer for the comment and we revised our table as below:

HOMA-IR index

total

Low

(<1.8)

Middle

(1.8~3.5)

High

(>3.5)

variable

N=396

n=129

n=136

n=131

P value

Leptin (ng/mL)

14.89

[8.48, 28.36]

9.96

[5.72, 16.15]

14.30

[7.81, 26.55]

23.30

[13.24, 41.21]

<0.001

HOMA-IR index

2.50

[1.50, 4.10]

1.20

[0.90, 1.50]

2.50

[2.10, 3.08]

5.10

[4.10, 7.70]

<0.001

Age (years)

63.72 ± 8.76

63.90 ± 9.71

63.89 ± 8.13

63.36 ± 8.45

0.620

FPG (mg/dL)

99.00

[89, 118.75]

91.00

[84.00, 97.00]

98.50

[90.25, 111.00]

118.00

[105.00, 145.00]

<0.001

Insulin (uU/mL)

9.89

[6.17, 14.93]

5.16

[3.96, 6.44]

9.99

[8.49, 11.89]

18.22

[13.91, 24.86]

<0.001

Uric Acid (mg/dL)

5.63 ± 1.52

5.25 ± 1.30

5.48 ± 1.39

6.18 ± 1.70

<0.001

ALT (U/L)

21.00

[16.00, 30.00]

18.00

[15.00, 26.50]

20.00

[15.00, 25.00]

26.00

[19.00, 38.00]

<0.001

Triglyceride (mg/dL)

118.00

[86.00, 165.00]

92.00

[73.00, 120.50]

125.50

[88.25, 167.75]

141.00

[106.00, 205.00]

<0.001

HDL-C (mg/dL)

52.00

[43.00, 61.00]

58.00

[50.00, 68.50]

51.00

[43.00, 59.75]

47.00

[40.00, 56.00]

<0.001

LDL-C (mg/dL)

109.69 ± 33.99

111.97 ± 32.56

113.38 ± 37.06

103.62 ± 31.38

0.047

Creatinine

(mg/dL)

0.80

[0.68, 0.97]

0.77

[0.66, 0.94]

0.82

[0.68, 1.00]

0.84

[0.73, 0.98]

0.027

Albumin

4.43 ± 0.25

4.43 ± 0.28

4.44 ± 0.26

4.42 ± 0.22

0.731

SBP (mmHg)

137.30 ± 17.49

134.58 ± 18.28

136.68 ± 15.96

140.63 ± 17.81

0.005

DBP (mmHg)

85.19 ± 10.98

82.38 ± 10.52

86.09 ± 11.06

87.02 ± 10.87

<0.001

BMI (kg/m2)

25.59± 3.84

23.46 ± 2.88

25.37 ± 3.05

27.92 ± 4.11

<0.001

WC (cm)

85.36 ± 10.83

82.81 ± 9.85

84.08 ± 10.00

89.17 ± 11.58

<0.001

Gender (male)

164 (41.4%)

42 (32.6%)

59 (43.4%)

63 (48.1%)

0.011

Smoking

50 (12.6%)

10 (7.8%)

17 (12.5%)

23 (17.6%)

0.017

Drinking

28 (7.1%)

8 (6.2%)

10 (7.4%)

10 (7.6%)

0.653

HTN

201 (50.8%)

43 (33.3%)

74 (54.4%)

84 (64.1%)

<0.001

DM

133 (33.6%)

25 (19.4%)

41 (30.1%)

67 (51.1%)

<0.001

Dyslipidemia

153 (38.6%)

36 (27.9%)

53 (39.0%)

64 (48.9%)

0.001

Question 6

-     The Authors in the Discussion section should not repeat the results from the previous section. Instead, the obtained results should be discussed and compared with the results of the previous studies related to this topic.

      Answer 6

      Thank you for the comments about this issue. We delete the repeat paragraph and added more content which related to previous study as below:

      Insulin is degraded by insulin-degrading enzymes in the liver [26]. The liver function has a profound impact on insulin degradation [27]. Indeed, there was a significant relationship be-tween HOMA-IR and ALT, as shown in Table 1, and Pearson’s correlation also indicated that HOMA-IR had a significant link with ALT, even after adjustment for age. However, HOMA-IR still maintained a significant relationship with serum leptin levels even after adjustment for ALT. Dehydration and hypoalbuminemia are often noted in older people [28] [29]. Additionally, serum albumin levels also serve as an important indicator of liver function [30]. In our study, the average serum albumin levels of each group were all within the normal range, and there was no relationship between albumin and HOMA-IR in our study. Because many low-molecular-weight proteins such as leptin are cleared by the kidneys, renal function plays a crucial role in the clearance of these proteins [31]. We took creatinine into consideration in model 3 for the linear regression, and HOMA-IR maintained a significant relationship with serum leptin levels.

      Reference :

  1. Farris, W., Mansourian, S., Chang, Y., Lindsley, L., Eckman, E. A., Frosch, M. P., ... & Guénette, S. (2003). Insulin-degrading enzyme regulates the levels of insulin, amyloid β-protein, and the β-amyloid precursor protein intracellular domain in vivo. Proceedings of the National Academy of Sciences, 100(7), 4162-4167.
  2. Borges, D. O., Patarrão, R. S., Ribeiro, R. T., de Oliveira, R. M., Duarte, N., Belew, G. D., ... & Macedo, M. P. (2021). Loss of postprandial insulin clearance control by Insulin-degrading enzyme drives dysmetabolism traits. Metabolism, 118, 154735.
  3. Hooper, L., Bunn, D., Jimoh, F. O., & Fairweather-Tait, S. J. (2014). Water-loss dehydration and aging. Mechanisms of Ageing and Development, 136, 50-58.
  4. Reuben, D. B., Moore, A. A., Damesyn, M., Keeler, E., Harrison, G. G., & Greendale, G. A. (1997). Correlates of hypoalbuminemia in community-dwelling older persons. The American journal of clinical nutrition, 66(1), 38-45.
  5. Moman, R. N., Gupta, N., & Varacallo, M. (2017). Physiology, albumin.
  6. Alwahsh, S. M., Xu, M., Seyhan, H. A., Ahmad, S., Mihm, S., Ramadori, G., & Schultze, F. C. (2014). Diet high in fructose leads to an overexpression of lipocalin-2 in rat fatty liver. World Journal of Gastroenterology: WJG, 20(7), 1807.

      Question 7

-     What are the limitations of the study?

Answer 7

We thank the reviewer for reminding us of this important issue. In response to this comment, we’ve added the limitation into the discussion: “However, there are some limitations to our study. We recruited participants from Northern Taiwan as the target population. These participants might differ from the general population, and selection bias should be considered; the findings cannot represent the entire middle-aged and elderly population in Taiwan. Although our study revealed obvious results, future studies using random sampling of communities with a wider range of regions should be considered. Moreover, other confounders, such as medication, hepatic steatosis, and other low-molecular-weight proteins, were not considered in our study. These potential confounders should be included in future studies.”

Reviewer 2 Report

The study is interesting and the relationship between leptin and insulin resistance among middle-aged and elderly populations in Taiwan is analyzed. According to the presented data, I agree with the conclusion that increased leptin levels are associated with high HOMA-IR, which might due to leptin resistance.

Nevertheless, the authors should check the data carefully: In table 1, the data of creatine and WC in total group were not reasonable. More information about a link between insulin resistance and other chronic disease should be introduced in introduction to help readers understand the study.

Author Response

Question 1

The study is interesting and the relationship between leptin and insulin resistance among middle-aged and elderly populations in Taiwan is analyzed. According to the presented data, I agree with the conclusion that increased leptin levels are associated with high HOMA-IR, which might due to leptin resistance.

Answer 1

We thank for reviewer’s comment

Question 2

Nevertheless, the authors should check the data carefully: In table 1, the data of creatine and WC in total group were not reasonable.

Answer 2

We apologize for the mistake we made and we have correct it. The corresponded content has edited and revised Table 1 is showed below:

HOMA-IR index

total

Low

(<1.8)

Middle

(1.8~3.5)

High

(>3.5)

variable

N=396

n=129

n=136

n=131

P value

Leptin (ng/mL)

14.89

[8.48, 28.36]

9.96

[5.72, 16.15]

14.30

[7.81, 26.55]

23.30

[13.24, 41.21]

<0.001

HOMA-IR index

2.50

[1.50, 4.10]

1.20

[0.90, 1.50]

2.50

[2.10, 3.08]

5.10

[4.10, 7.70]

<0.001

Age (years)

63.72 ± 8.76

63.90 ± 9.71

63.89 ± 8.13

63.36 ± 8.45

0.620

FPG (mg/dL)

99.00

[89, 118.75]

91.00

[84.00, 97.00]

98.50

[90.25, 111.00]

118.00

[105.00, 145.00]

<0.001

Insulin (uU/mL)

9.89

[6.17, 14.93]

5.16

[3.96, 6.44]

9.99

[8.49, 11.89]

18.22

[13.91, 24.86]

<0.001

Uric Acid (mg/dL)

5.63 ± 1.52

5.25 ± 1.30

5.48 ± 1.39

6.18 ± 1.70

<0.001

ALT (U/L)

21.00

[16.00, 30.00]

18.00

[15.00, 26.50]

20.00

[15.00, 25.00]

26.00

[19.00, 38.00]

<0.001

Triglyceride (mg/dL)

118.00

[86.00, 165.00]

92.00

[73.00, 120.50]

125.50

[88.25, 167.75]

141.00

[106.00, 205.00]

<0.001

HDL-C (mg/dL)

52.00

[43.00, 61.00]

58.00

[50.00, 68.50]

51.00

[43.00, 59.75]

47.00

[40.00, 56.00]

<0.001

LDL-C (mg/dL)

109.69 ± 33.99

111.97 ± 32.56

113.38 ± 37.06

103.62 ± 31.38

0.047

Creatinine

(mg/dL)

0.80

[0.68, 0.97]

0.77

[0.66, 0.94]

0.82

[0.68, 1.00]

0.84

[0.73, 0.98]

0.027

Albumin

4.43 ± 0.25

4.43 ± 0.28

4.44 ± 0.26

4.42 ± 0.22

0.731

SBP (mmHg)

137.30 ± 17.49

134.58 ± 18.28

136.68 ± 15.96

140.63 ± 17.81

0.005

DBP (mmHg)

85.19 ± 10.98

82.38 ± 10.52

86.09 ± 11.06

87.02 ± 10.87

<0.001

BMI (kg/m2)

25.59± 3.84

23.46 ± 2.88

25.37 ± 3.05

27.92 ± 4.11

<0.001

WC (cm)

85.36 ± 10.83

82.81 ± 9.85

84.08 ± 10.00

89.17 ± 11.58

<0.001

Gender (male)

164 (41.4%)

42 (32.6%)

59 (43.4%)

63 (48.1%)

0.011

Smoking

50 (12.6%)

10 (7.8%)

17 (12.5%)

23 (17.6%)

0.017

Drinking

28 (7.1%)

8 (6.2%)

10 (7.4%)

10 (7.6%)

0.653

HTN

201 (50.8%)

43 (33.3%)

74 (54.4%)

84 (64.1%)

<0.001

DM

133 (33.6%)

25 (19.4%)

41 (30.1%)

67 (51.1%)

<0.001

Dyslipidemia

153 (38.6%)

36 (27.9%)

53 (39.0%)

64 (48.9%)

0.001

Question 3

More information about a link between insulin resistance and other chronic disease should be introduced in introduction to help readers understand the study.

Answer 3

Thank you for the accurate comments about this issue. We revised the Introduction as below:

“Insulin resistance relates to the accumulation of adipocytes in metabolic disorders, although the mechanism is not fully understood. The large amounts of data from previous hu-man studies and animal research indicate a positive relationship between insulin resistance and metabolic disorders [1]. Furthermore, insulin resistance serves as the key feature of these metabolic disorders, such as diabetes mellitus (DM). Insulin resistance decreases the sensitivity of the physiological function of insulin and leads to type 2 DM [2] [3]. DM-related complications carry serious consequences, including nephropathy, retinopathy, peripheral neuropathy, end-stage renal disease, cardiovascular disease, cerebrovascular disease, and diabetic peripheral circulatory disorder [4] [5] [6] [7] [8]. Recent studies even indicated that insulin resistance relates to other diseases, such as hypertension, cancer, and autoimmune diseases [9] [10] [11]. Diseases related to insulin resistance have created a significant socioeconomic bur-den for patients and society.

Leptin is believed to act as a negative feedback signal that regulates the energy balance by inhibiting food intake, and it reduces fat storage [12]. Leptin is mainly produced by adipocytes, and the plasma leptin level correlates with the number of adipocytes in the body [13]. Excessive fat tissue does not merely elevate plasma leptin levels but relates to insulin resistance [14], which is a crucial feature of metabolic disorders. Moreover, leptin resistance was also found in the obese population [15], who are vulnerable to insulin resistance. Hence, we aimed to investigate the relationship between leptin levels and insulin resistance in the population aged between 50 and 85 years in Taiwan. The results of our study may provide a possible reference for insulin resistance in primary care. ''

Reference:

  1. Blüher, Matthias. "Adipose tissue inflammation: a cause or consequence of obesity-related insulin resistance?." Clinical science 130.18 (2016): 1603-1614.
  2. Freeman AM, Pennings N. Insulin Resistance. 2021 Jul 10. In: StatPearls [Internet]. Treasure Island (FL): StatPearls Publishing; 2021 Jan–. PMID: 29939616.
  3. Kahn BB, Flier JS. Obesity and insulin resistance. J Clin Invest. 2000 Aug;106(4):473-81. doi: 10.1172/JCI10842. PMID: 10953022; PMCID: PMC380258.
  4. Ferrannini E, Natali A, Capaldo B, Lehtovirta M, Jacob S, Yki-Järvinen H. Insulin resistance, hyperinsulinemia, and blood pressure: role of age and obesity. European Group for the Study of Insulin Resistance (EGIR). Hypertension. 1997 Nov;30(5):1144-9. doi: 10.1161/01.hyp.30.5.1144. PMID: 9369268.
  5. L.N. Tseng, Y.H. Tseng, Y.D. Jiang, C.H. Chang, C.H. Chung, B.J. Lin, et al. Prevalence of hypertension and dyslipidemia and their associations with micro- and macrovascular diseases in patients with diabetes in Taiwan: an analysis of nationwide data for 2000-2009. J Formos Med Assoc, 111 (11) (2012), pp. 625-636
  6. H.Y. Li, Y.D. Jiang, C.H. Chang, C.H. Chung, B.J. Lin, L.M. Chuang. Mortality trends in patients with diabetes in Taiwan: a nationwide survey in 2000-2009. J Formos Med Assoc, 111 (11) (2012), pp. 645-650
  7. Y.J. Sheen, T.C. Li, J.L. Lin, W.C. Tsai, C.D. Kao, C.T. Bau, et al. Association between thermal threshold abnormalities and peripheral artery disease in patients with type 2 diabetes. Medicine (Baltimore), 97 (51) (2018), Article e13803
  8. Y.J. Sheen, W.H. Sheu. Association between hypoglycemia and dementia in patients with type 2 diabetes. Diabetes Res Clin Pract, 116 (2016), pp. 279-287
  9. C.L. Hsu, W.H. Sheu. Diabetes and shoulder disorders. J Diabetes Investig, 7 (5) (2016), pp. 649-651
  10. Orgel E, Mittelman SD. The links between insulin resistance, diabetes, and cancer. Curr Diab Rep. 2013 Apr;13(2):213-22. doi: 10.1007/s11892-012-0356-6. PMID: 23271574; PMCID: PMC3595327.
  11. Chung CP, Oeser A, Solus JF, Gebretsadik T, Shintani A, Avalos I, Sokka T, Raggi P, Pincus T, Stein CM. Inflammation-associated insulin resistance: differential effects in rheumatoid arthritis and systemic lupus erythematosus define potential mechanisms. Arthritis Rheum. 2008 Jul;58(7):2105-12. doi: 10.1002/art.23600. PMID: 18576352; PMCID: PMC2755593.
  12. Kelesidis T, Kelesidis I, Chou S, Mantzoros CS. Narrative review: the role of leptin in human physiology: emerging clinical applications. Ann Intern Med. 2010 Jan 19;152(2):93-100. doi: 10.7326/0003-4819-152-2-201001190-00008. PMID: 20083828; PMCID: PMC2829242.
  13. Schwartz, Michael W. "Central nervous system regulation of food intake." Obesity 14.2S (2006): 1S.
  14. Osegbe I, Okpara H, Azinge E. Relationship between serum leptin and insulin resistance among obese Nigerian women. Ann Afr Med. 2016 Jan-Mar;15(1):14-9. doi: 10.4103/1596-3519.158524. PMID: 26857932; PMCID: PMC5452686.
  15. Smith U, Kahn BB. Adipose tissue regulates insulin sensitivity: role of adipogenesis, de novo lipogenesis and novel lipids. J Intern Med. 2016 Nov;280(5):465-475. doi: 10.1111/joim.12540. Epub 2016 Oct 3. PMID: 27699898; PMCID: PMC5218584.
  16. Enriori, P. J., Evans, A. E., Sinnayah, P., & Cowley, M. A. (2006). Leptin resistance and obesity. Obesity14(S8), 254S-258S.

Reviewer 3 Report

This is an  observational study on a population of 396 middle-aged persons in Taiwan.Insulin resistance and several other markers including leptin were measured.Authors found hat the high HOMA-IR group was more likely to have higher leptin level (105 of the results).

Comments:

1.First it has to be appreciated that authors mentioned the number of the application of the study protocol.

2.this stuy is of importance because it gives the opportunity to have data from chinese people with western style nutrition(?).The mean body mass index(BMI) should be higher than that of the mainland chinese population.

If this is true it should be mentioned because it may have epidemiological importance.

3.The increased ALT-level may indicate the presence of NASH.Ultrasound informations would be appreciated/discussion of this point should be presented.It should also be discussed that increase of insulin serum level can also be contributed by reduced clearance in the steatotic liver.

4.older people may develop some dehydration and hypoalbuminemia.Albumin serum level should be given.

 5.  as leptin like other low molecular weight proteins(Alwhash S et al World J Gastroenterol ( 2014 Feb 21;20(7):1807-21. doi: 10.3748/wjg.v20.i7.1807) is cleared by the kidney.Its serum level may increase by reduced kidney function.Therefore the use  leptin/creatinine ratio may be appropriate.

6.mesaurement of other low molecular weight proteins like lipocalin-2 or hepcidin in this special population may be of interest.

Author Response

Question 1

1.First it has to be appreciated that authors mentioned the number of the application of the study protocol.

Answer 1

We thank for reviewer’s comment

Question 2

2.this stuy is of importance because it gives the opportunity to have data from chinese people with western style nutrition(?).The mean body mass index(BMI) should be higher than that of the mainland chinese population. If this is true it should be mentioned because it may have epidemiological importance.

Answer 2

We want to thank the reviewer for constructive and insightful comments. After we study on previous research about the Chinese population, we found out that the elder population in Taiwan has higher BMI than the elder population in China. Hence, we added the finding in the discussion below:

“Our study also has epidemiology importance. Due to the development of China, the population of China has healthier lifestyle and has longer longevity than before [42]. The obesity of the elder population emerged as health problem [43]. When we compared China population with the Taiwan population in our study, the indicators of metabolic syndrome like BMI and rate of DM are higher in Taiwan's elder population than in China’s counterpart. Our study can provide insight into the future elder population of China.”

Reference:

  1. Wang, L., Li, Y., Li, H., Holdaway, J., Hao, Z., Wang, W., & Krafft, T. (2016). Regional aging and longevity character-istics in China. Archives of gerontology and geriatrics, 67, 153-159.
  2. Salihu, H. M., Bonnema, S. M., & Alio, A. P. (2009). Obesity: what is an elderly population growing into?. Maturitas, 63(1), 7-12. “Third, our study has epidemiology importance to provide possible prediction of future elder population of China”

Question 3

3.The increased ALT-level may indicate the presence of NASH.Ultrasound informations would be appreciated/discussion of this point should be presented.It should also be discussed that increase of insulin serum level can also be contributed by reduced clearance in the steatotic liver.

Answer 3

Thank you for this valuable comment. We did not have the equipment to perform abdominal ultrasounds in every local medical clinic in this study, so we can not present ultrasound information in our article. Regarding liver function, model 3 adjustment of linear regression also included alanine aminotransferase (ALT), and the result still showed significant relationship between HOMA-IR and serum leptin level. We describe this finding and add this into the discussion “Insulin is degraded by insulin-degrading enzymes in the liver [26]. The liver function has a profound impact on insulin degradation [27]. Indeed, there was a significant relationship be-tween HOMA-IR and ALT, as shown in Table 1, and Pearson’s correlation also indicated that HOMA-IR had a significant link with ALT, even after adjustment for age. However, HOMA-IR still maintained a significant relationship with serum leptin levels even after adjustment for ALT.”

Reference:

  1. Farris, W., Mansourian, S., Chang, Y., Lindsley, L., Eckman, E. A., Frosch, M. P., ... & Guénette, S. (2003). Insulin-degrading enzyme regulates the levels of insulin, amyloid β-protein, and the β-amyloid precursor protein intracellular domain in vivo. Proceedings of the National Academy of Sciences, 100(7), 4162-4167.
  2. Borges, D. O., Patarrão, R. S., Ribeiro, R. T., de Oliveira, R. M., Duarte, N., Belew, G. D., ... & Macedo, M. P. (2021). Loss of postprandial insulin clearance control by Insulin-degrading enzyme drives dysmetabolism traits. Metabolism, 118, 154735.

“Moreover, other confounders, like medication, hepatic steatosis, and other low molecular weight proteins, were not collected in our study. Those potential confounders should be included in future studies.”

Question 4

4.older people may develop some dehydration and hypoalbuminemia. Albumin serum level should be given.

Answer 4

We appreciate the reviewer’s comment. Albumin data was analyzed and added to Table 1. The average serum albumin level of the low HOMA-IR group, middle HOMA-IR group, and high HOMA-IR group were 4.43, 4.44, and 4.42, respectively. The serum albumin levels in each group were all in the normal range, and there was no relationship between albumin and HOMA-IR in our study. We also add this finding in our discussion “Dehydration and hypoalbuminemia are often noted in older people [28] [29]. Additionally, serum albumin levels also serve as an important indicator of liver function [30]. In our study, the average serum albumin levels of each group were all within the normal range, and there was no relationship between albumin and HOMA-IR in our study.”

Refernce:

  1. Hooper, L., Bunn, D., Jimoh, F. O., & Fairweather-Tait, S. J. (2014). Water-loss dehydration and aging. Mechanisms of Ageing and Development136, 50-58.
  2. Reuben, D. B., Moore, A. A., Damesyn, M., Keeler, E., Harrison, G. G., & Greendale, G. A. (1997). Correlates of hypoalbuminemia in community-dwelling older persons. The American journal of clinical nutrition66(1), 38-45.
  3. Moman, R. N., Gupta, N., & Varacallo, M. (2017). Physiology, albumin.

Question 5

  1. as leptin like other low molecular weight proteins(Alwhash S et al World J Gastroenterol ( 2014 Feb 21;20(7):1807-21. doi: 10.3748/wjg.v20.i7.1807) is cleared by the kidney. Its serum level may increase by reduced kidney function.Therefore the use leptin/creatinine ratio may be appropriate.

Answer 5

We thank the reviewer for reminding us of this issue. Indeed, renal function plays an important role in leptin clearance. We put creatinine into consideration in model 3 of linear regression, and HOMA-IR still maintained significant relationship with serum leptin level. We added the description in the discussion below: “Because many low-molecular-weight proteins such as leptin are cleared by the kidneys, renal function plays a crucial role in the clearance of these proteins [31]. We took creatinine into consideration in model 3 for the linear regression, and HOMA-IR maintained a significant relationship with serum leptin levels.”

Reference:

  1. Alwahsh, S. M., Xu, M., Seyhan, H. A., Ahmad, S., Mihm, S., Ramadori, G., & Schultze, F. C. (2014). Diet high in fructose leads to an overexpression of lipocalin-2 in rat fatty liver. World Journal of Gastroenterology: WJG, 20(7), 1807.

Question 6

6.mesaurement of other low molecular weight proteins like lipocalin-2 or hepcidin in this special population may be of interest.

Answer 6

We thank the reviewer for reminding us of this important issue. Regretfully, our study did not included other low molecular weight proteins like lipocalin-2 and hepacidin. We will consider low molecular weight proteins in our future study. In response to this comment, we’ve added this limitation into the discussion: “Moreover, other confounders, such as medication, hepatic steatosis, and other low-molecular-weight proteins, were not considered in our study. These potential confounders should be included in future studies.”

Round 2

Reviewer 1 Report

The Authors have made corrections according to the Reviewer's suggestions.

Author Response

We appreciate you for your constructive comments which strengthen our article. Thank you for your time.

Reviewer 3 Report

Authors have invested efforts in improving the quality of the manuscript.

I still consider that the leptin/creatinine ratio instead of leptin and creatinin alone would be appropriate

Author Response

Question:

I still consider that the leptin/creatinine ratio instead of leptin and creatinin alone would be appropriate

Answer:

Thank reviewer for your concern. In order to evaluate the leptin-creatinine ratio, we reviewed our raw data and analyzed it again. The result showed a significantly positive relationship between leptin-creatinine ratio and HOMA-IR in linear regression through all three models. The result was shown as Table 4, and related content was also shown below:

Table 4. Linear regression between leptin-creatinine ratio and HOMA-IR index

Model 1

Model 2

Model 3

β

S.E.

P value

β

S.E.

P value

β

S.E.

P value

Leptin

0.037

0.010

<0.001

0.031

0.014

0.028

0.041

0.013

0.001

Model 1 was unadjusted;

Model 2 was adjusted for age, gender, smoking, drinking, and BMI;

Model 3 was adjusted for age, sex, smoking, drinking, BMI, triglyceride, systolic blood pressure, fasting plasma glucose, uric acid, and ALT level

“Furthermore, the leptin-creatinine ratio also showed significantly positive relationship with HOMA-IR in the same multivariate linear regression model.” (in Abstract)

“Then, the same linear regression model was performed again to evaluate the relationship between leptin-creatinine ratio and insulin resistance.” (in Materials and Methods)

“We used the same multivariate model in Table 3 to evaluate the relationship between the leptin-creatinine ratio in Table 4. The relationships still remained significantly positive with beta coefficient of 0.037 in model 1, 0.031 in model 2, and 0.041 in model 3” (in Results)

“Because many low-molecular-weight proteins such as leptin are cleared by the kidneys, renal function plays a crucial role in the clearance of these proteins [34]. Creatinine was taken into consideration further in Table 4 for the linear regression. In Table 4, we used the same linear regression model in Table 3 to evaluate the relationship between the leptin-creatinine ratio and HOMA-IR IN in table 4. The result revealed significantly positive relationship in all three models. Linear regression analysis confirmed that leptin was an independent risk factor for insulin resistance even considering creatinine level.” (in Discussion)

Reference :

  1. Alwahsh, S. M., Xu, M., Seyhan, H. A., Ahmad, S., Mihm, S., Ramadori, G., & Schultze, F. C. (2014). Diet high in fructose leads to an overexpression of lipocalin-2 in rat fatty liver. World Journal of Gastroenterology: WJG, 20(7), 1807.
